# Influence of Chitin Nanocrystals on the Crystallinity and Mechanical Properties of Poly(hydroxybutyrate) Biopolymer

**DOI:** 10.3390/polym14030562

**Published:** 2022-01-29

**Authors:** Marta Zaccone, Mitul Kumar Patel, Laurens De Brauwer, Rakesh Nair, Maria Luana Montalbano, Marco Monti, Kristiina Oksman

**Affiliations:** 1Proplast, Via Roberto di Ferro 86, 15122 Alessandria, Italy; luana.montalbano@proplast.it (M.L.M.); marco.monti@proplast.it (M.M.); 2Division of Materials Science, Department on Engineering Sciences and Mathematics, Luleå University of Technology, SE 97187 Luleå, Sweden; mitul.kumar.patel@ltu.se; 3Bio Base Europe Pilot Plant (BBEPP), Rodenhuizekaai 1, 9042 Gent, Belgium; laurens.de.brauwer@bbeu.org (L.D.B.); rakesh.nair@bbeu.org (R.N.); 4Mechanical & Industrial Engineering, University of Toronto, Toronto, ON M5S 3BS, Canada; 5Wallenberg Wood Science Center (WWSC), Luleå University of Technology, SE 97187 Luleå, Sweden

**Keywords:** biopolymer, polyhydroxy butyrate, chitin nanocrystals, nanocomposites, crystallinity, mechanical properties

## Abstract

This study focuses on the use of pilot-scale produced polyhydroxy butyrate (PHB) biopolymer and chitin nanocrystals (ChNCs) in two different concentrated (1 and 5 wt.%) nanocomposites. The nanocomposites were compounded using a twin-screw extruder and calendered into sheets. The crystallization was studied using polarized optical microscopy and differential scanning calorimetry, the thermal properties were studied using thermogravimetric analysis, the viscosity was studied using a shear rheometer, the mechanical properties were studied using conventional tensile testing, and the morphology of the prepared material was studied using optical microscopy and scanning electron microscopy. The results showed that the addition of ChNCs significantly affected the crystallization of PHB, resulting in slower crystallization, lower overall crystallinity, and smaller crystal size. Furthermore, the addition of ChNCs resulted in increased viscosity in the final formulations. The calendering process resulted in slightly aligned sheets and the nanocomposites with 5 wt.% ChNCs evaluated along the machine direction showed the highest mechanical properties, the strength increased from 24 to 33 MPa, while the transversal direction with lower initial strength at 14 MPa was improved to 21 MPa.

## 1. Introduction

In recent years, bio-based and biodegradable polymers, such as polylactic acid (PLA), polycaprolactone (PCL), and thermoplastic starch-based polymers (TPS), have emerged as promising alternatives to fossil-based materials because of their many advantages, such as biodegradability, environmental compatibility, and their renewable origin [1,2]. In addition to these biopolymers, polyhydroxyalkanoates (PHAs) also possess these properties [3,4,5,6]. These bio-polyesters are synthesized by bacterial species and have shown promise in many applications, such as biomedical and pharmaceutical, agricultural, single-use products, and packaging [4,7,8]. Among the family of PHAs, poly(3-hydroxybutyrate) (PHB) is one of the most extensively studied because of its physical and mechanical properties, which are comparable to those of petrochemical-derived polymers like polypropylene (PP) [9,10,11,12,13]. Indeed, PHB exhibits good mechanical and oxygen barrier properties. Moreover, it is stable under normal usage conditions but undergoes rapid biodegradation under composting conditions [14]. Nevertheless, PHB is susceptible to thermal degradation, has a narrow processing window, and shows severe post-process embrittlement, which reduces the applicability of this biopolymer [3,10,15,16,17].

Several solutions have been presented to overcome these shortcomings [18]. One possibility is the synthesis of copolymers like poly hydroxybutyrate-co-valerate (PHBV). Indeed, a stiffer product with a higher HB content or a tougher material with a higher HV content can be obtained by adjusting the hydroxybutyrate/hydroxyvalerate (HB/HV) ratio. Nonetheless, this approach is not cost-effective, and the presence of the copolymer affects the crystallization kinetics of PHB, resulting in longer processing cycle times [19]. Another alternative is to reduce the brittleness of PHB by performing annealing, which substantially improves the processability and mechanical stability of the polymer [20], but this also leads to longer processing cycles. Finally, a third possibility is the formulation of nanocomposites, as nanosized additives may enhance the final material properties [21,22]. Many studies have focused on PHB-or PHBV/clay-based nanocomposites [1,23,24,25,26,27]. Similarly, cellulose nanowhiskers (CNWs) are of high interest for biopolymers because they can be obtained from renewable sources and can improve the mechanical properties of the final nanocomposites [28]. Another way to improve the mechanical properties of biopolymers is to alter the molecular orientation of the polymer. Oriented polymer films and tapes can be obtained by melt-drawing processes, such as film (sheet) calendering and blow molding. During melt drawing, the polymer molecules are stretched at the exit of the extruder die and are oriented along the flow direction.

Chitin is an abundant natural polymer that has attracted considerable attention from the scientific community [29,30,31]. Chitin can be extracted from the exoskeleton of crustaceans, such as crabs and shrimp, and is used in a variety of applications owing to its biodegradability and biocompatibility [32,33]. Moreover, owing to the hierarchical structure of chitin, chitin nanofibers (ChNFs) and nanocrystals (ChNCs) can be extracted from chitin in a top-down manner. These chitin nanomaterials have a high aspect ratio, high specific surface area, and impressive mechanical properties [30,31,34,35,36]. There are several reports on the lab-scale preparation of ChNCs from different raw material sources, such as squid pen [36], crab shells [37,38], and shrimp shells [39,40], using conventional hydrolysis in HCl solution. The isolated nanocrystals had aspect ratios of 10–55 and have been studied as nanocomposite reinforcement materials. Hydrolysis with a strong acid (H_2_SO_4_) for ChNC production has only been reported a few times [41,42]. Furthermore, to the best of our knowledge, the large-scale production of ChNCs has not been explored.

Several papers have been published on the use of chitin nanomaterials in biopolymer nanocomposites [31,32,43,44,45,46,47]. Li et al. [31] studied the nucleation ability of ChNCs on PHBs. They used ChNCs with and without surface treatment and found that chitin nanocrystals without any modification showed good nucleation ability. Singh et al. [45] studied PLA-ChNC nanocomposites and showed that even a small amount of ChNCs increased the crystallization rate, barrier performance, and hydrolytic degradation rate of PLA. Furthermore, Scaffaro et al. discussed the preparation and use of ChNCs in combination with cellulose nanocrystals in the PLA matrix and presented many studies on cellulose nanocomposites in a review article [43].

However, to the best of our knowledge, only one study has been published on ChNC nanocomposites with PHB as the matrix. In this study, we investigated the feasibility of pilot-scale produced PHB and ChNCs for the preparation of nanocomposites using melt extrusion and sheet calendering under industrially feasible processing conditions. We studied how the addition of ChNCs affected the crystallization behavior, thermal, rheological, and mechanical properties of PHB.

## 2. Materials and Methods

### 2.1. Production of PHB

PHB was produced via fed-batch fermentation of *Paraburkholderia sacchari* at 29 °C in a 1500 L bioreactor according to the method described by Kim et al. [48] with some modifications. Instead of ammonium limitation, as proposed by Kim et al., phosphate limitation was used for PHB production. A total of 1 mL of *P. sacchari* culture was thawed from the working glycerol stock and inoculated into a 500 mL sterile mineral medium flask. The shake flask was incubated for 20 h at 29 °C and 200 rpm, after which the bacteria were inoculated into the bioreactor for growth. The organism was grown in a mineral medium containing glucose, and automated fed-batch culture fermentation was carried out with the use of concentrated glucose shots (600 g/L) when the pO_2_ exceeded 20% of the set value, as shown in Appendix A.

The glucose concentration of the culture broth was maintained at 10–20 g/L based on the pO_2_ value. The final cell concentration, PHB concentration, and PHB productivity increased with increasing cell concentrations and the maximum PHB content (79 g/L) and total cell dry weight (111 g/L) were obtained at the end of the fermentation process. The maximum PHB content, amounting to 74% of the dry cell weight, and a productivity of 1.6 g/L·h at a yield of 0.2 kg PHB/kg glucose was achieved at the end of fermentation (Appendix A).

The biomass at the end of the fermentation process was harvested and processed to purify the intracellular PHB. The biomass first underwent cell lysis under high-pressure homogenization, followed by enzymatic hydrolysis of the cellular material, which resulted in the recovery of intact PHB granules. After washing with water to remove traces of salts and contaminants, PHB was recovered and dried in a vacuum tray. Based on thermogravimetric analysis (TGA), the PHB had a recovery yield of 90% with a purity exceeding 95%. The biopolymer was stored at 4 °C until use.

### 2.2. Production of ChNCs

ChNCs were produced by diluting chitin powder from shrimp shells with an average molecular weight of 203 g/mol (Glentham Life Sciences Ltd., Corsham, UK) in 35 wt.% H_2_SO_4_ (Brenntag NV, Deerlijk, Belgium) for 2 h at 60 °C in a Pfaudler AE 400 glass-lined reactor (Thaletec GmbH, Thale, Germany). The liquid was neutralized with 30% NaOH and subsequently diafiltered with 0.01 *v*/*v*% acetic acid (Brenntag NV, Deerlijk, Belgium) to reach a conductivity of 130 µS/cm. Unhydrolyzed chitin was sedimented on a Clara 20 disc-stack centrifuge (Alfa Laval, Lund, Sweden), and the ChNC-containing supernatant was concentrated using a Carl Canzler Wiped Film evaporator (Quadrant EPP, Tielt Belgium). The concentrate was freeze-dried to obtain a dry, stable product. The process scheme to produce ChNCs is shown in Appendix A. The ChNCs were stored at 4 °C until mixing. About 1.4 kg of ChNCs was produced, with a process yield of approximately 34 wt.%.

### 2.3. Compounding and Sheet Calendering

PHB-ChNC nanocomposites were prepared in a co-rotating twin-screw extruder (Leistritz 27E, Nuremberg, Germany), with a 27 mm screw diameter and a length-to-diameter ratio L/D of 40. The screw speed was maintained at 300 rpm for all materials, and the temperature profile was set at 165–175 °C, with the temperature being the highest at the die. Vacuum venting was used to remove air and moisture. The freeze-dried ChNC powder was fed into the melt PHB using a gravimetric side feeder (Brabender, Duisburg, Germany). Nanocomposites with two different ChNC contents (1 and 5 wt.%) were prepared. Figure 1 shows the screw profile and process layout.

The obtained nanocomposites were extruded into sheets using a single-screw extruder (BGplast SD30, B.G. Plast Impianti SRL, Marnate, Italy) equipped with a flat die (width 200 mm) and a calendering system (Dr. Collins CR136/350, Collin Lab and pilot solutions, Maitenbeth, Germany). The prepared compounds were dried before extrusion at 80 °C for 6 h. The extruded sheets were approximately 500 µm thick. The temperature profile was set at 180–190 °C. The temperature of the rolls was set at 30 °C, and the speed was 0.8 mm/min. Figure 2 shows the used process layout and calendered sheets of PHB nanocomposite.

### 2.4. Characterization

The morphology of the ChNCs before and after freeze-drying was studied using an optical microscope (Nikon Eclipse LV100 Pol; Bergman Labora AB, Danderyd, Sweden). The ChNC dispersion was diluted in distilled water to a concentration of 0.1 wt.%, followed by magnetic stirring for 2 h before optical microscopy. In addition, a flow birefringence study was performed to evaluate the quality of the nanocrystals and the effect of freeze-drying on the dispersion of the nanocrystals. The diluted dispersions (0.5 wt.%) were placed between cross-polarized filters and photographed.

The thermal properties of the prepared PHB, ChNCs, and nanocomposites were studied by TGA (Q500 TA Instruments, New Castle, DE, USA). TGA was performed in a nitrogen atmosphere at a heating rate of 10 °C/min (gas flow rate of 60 mL/min) at 50–800 °C.

The crystallization rate and morphology of neat PHB and the nanocomposites were studied using polarized optical microscopy (POM, Nikon Eclipse LV100 Pol, Bergman Labora AB, Danderyd, Sweden) equipped with a Linkam THM600 (Tadworth, UK) hot stage and a charge-coupled device camera. PHB was melted between two glass covers at 200 °C and cooled to room temperature, and crystallization and crystal nucleation were recorded.

Differential scanning calorimetry (DSC) was performed on a Q800 instrument (TA Instruments, New Castle, DE, USA). Approx. 5.5 g of each sample was used for DSC. The first heating scan was performed to eliminate the thermal history of the tested materials. A cooling scan from 210 to −10 °C and a second heating scan from −10 to 210 °C, both at 10 °C/min rate. The degree of crystallinity was calculated according to Equation (1):(1)$$X_{c}\left[\%\right]=\frac{\left({\Delta H}_{m}-{\;\Delta H}_{\mathrm{cc}}\right)}{\left(\frac{100\;-\;\mathrm{wt}.\%}{100}\right)}\times \frac{100}{\Delta H_{0}}$$
where X_c_ is the crystalline fraction of the matrix, ΔH_m_ is the melting enthalpy (J/g), ΔH_cc_ is the cold crystallization enthalpy (J/g), wt.% is the ChNC content by weight, and ΔH_0_ is the theoretical crystallization enthalpy of 100% crystalline PHB. In this calculation, 146 J/g was used as the value of ΔH_0_ [49].

The mechanical properties were studied using a conventional tensile tester (Zwick Roell Z010 model, Ulm, Germany), and the test was performed according to the UNI EN ISO 527-3 standard. The test was performed in the machine and transfer direction of the calendered sheets (MD and TD). The load cell had a maximum capacity of 10 kN. The samples (Type 2, according to the standard) had a rectangular shape, with a total length of over 150 mm, a width of 25 mm, and a thickness of 500 µm. The crosshead speed was 5 mm/min and the applied preload was 2 MPa. The strain was calculated by dividing the change in length by the initial length of the sample (ε = ∆L/L).

Rheological characterization was performed in a nitrogen atmosphere using a strain-controlled rotational rheometer (Rheometric Scientific ARES model 2KFRT, TA Instruments, New Castle, DE, USA) with a parallel-plate geometry (25 mm diameter). The purpose of rheological characterization was to understand how the presence of ChNCs affects the rheological behavior of neat PHB. Test specimens were cut from extruded sheets (500 µm thickness). Frequency sweep tests were performed in the 0.1–100 rad/s frequency range, with a fixed strain of 10%. The test temperature was set at 190 °C.

The morphologies of neat PHB and its nanocomposites were investigated using optical microscopy (OM) and scanning electron microscopy (SEM). OM was performed using a Nikon Eclipse LV100N Pol (Bergman Labora AB, Danderyd, Sweden) and SEM was performed using a JEOL JSM-6460LV (JEOL, Tokyo, Japan). The specimens were cryo-fractured in liquid nitrogen and coated with a thin layer (13 nm) of platinum before observation using an EM ACE200 (Leica vacuum coater, Wetzlar, Germany).

## 3. Results

Figure 3 shows the microstructure and birefringence of ChNCs in water dispersions before and after freeze-drying. In the micrograph of never-dried ChNCs, no ChNCs can be seen, indicating successful isolation of chitin into the nanocrystals, see Figure 3a. This is an expected result because the size of the crystals is below the OM resolution limit. In addition, the produced ChNCs exhibited a typical birefringence pattern between the cross-polarized filters, as shown in Figure 3b, because of the chiral nematic liquid crystalline phase in equilibrium with the isotropic phase. Furthermore, the AFM height image in Appendix A confirms the presence of nanocrystals before the freeze-drying step. Figure 3c shows freeze-dried ChNCs redispersed in water, and the micrograph shows that freeze-drying results in micrometer-sized flake-like particles and loss of birefringence, as shown in Figure 3d. These results show that the drying step makes it more difficult for ChNCs to be redispersed in water by forming irreversible strong hydrogen bonds, resulting in agglomeration and loss of birefringence.

Figure 4 shows the TGA results of the ChNCs, neat PHB, and their nanocomposites. In the case of ChNCs, the first weight loss was obtained at 40–100 °C, which is attributed to the presence of residual humidity. The main weight loss was observed in the 250–400 °C range, which corresponds to the thermal decomposition of the polysaccharide structure [50]. The final residue of the ChNCs at 800 °C was approximately 25 wt.%, owing to the presence of sulfate groups on the surface of the ChNCs, which were introduced during acid hydrolysis and could act as a flame retardant, as reported by Roman and Winter for cellulose nanocrystals [51]. Neat PHB and both nanocomposites showed one main thermal degradation step at approximately 275 °C. This value corresponds to the Ton-set of the materials, which indicates the temperature at which the polymer exhibits a weight loss of 95 wt.% of its initial weight. No differences were observed between neat PHB and the produced nanocomposites with 1 and 5 wt.% ChNCs, except for the presence of a second small shoulder at approximately 350 °C, which is attributable to the presence of ChNCs. The increasing amount of ChNCs in the matrix corresponds to the increasing height of the second peak.

Figure 5 shows the crystallization of PHB and its ChNC nanocomposites at (a) 80 °C after 1 min and (b) 60 °C after 2 min. From Figure 5(a1), it is evident that neat PHB had already started to crystallize and formed spherulites at 80 °C during the first minute, whereas the nanocomposites showed no spherulite development at this temperature during the first minute, shown in Figure 5(a2,a3). This indicates that PHB has a higher crystallization rate than the ChNCs. Figure 5(b1–b3) was taken at a lower temperature, 60 °C, and after 2 min, neat PHB and both nanocomposites showed spherulite formation, as shown in Figure 5(b1–b3). Larger spherulites in the form of helical strands radiating from a nucleation point were observed for neat PHB. On the other hand, several smaller spherulites were observed for PHB/1ChNC and PHB/5ChNC, with the size of the spherulites decreasing as the quantity of ChNCs in the nanocomposites increased. This phenomenon is attributable to the higher number of nucleation sites provided by ChNCs, leading to a larger number of spherulites, which in turn limits the ability of the spherulites to grow in size [52]. Furthermore, spherulites could not form in specific areas of nanocomposites, resulting in voids. This phenomenon is caused by a reduction in polymer availability in the particular area owing to the presence of agglomerated ChNCs. PHB/5ChNC exhibited the largest number of voids.

Table 1 shows the DSC results and degree of crystallinity of neat PHB and the produced nanocomposites. Based on the results, it is evident that the degree of crystallinity was lower in the presence of ChNCs. Figure 6a shows the DSC results pertaining to the cooling cycles of neat PHB and its nanocomposites. The graph of the cooling cycle and the crystallization temperatures (Tc) values in Table 1 show that the Tc of neat PHB is 66 °C, and the Tc shifts to lower temperatures on increasing the ChNC content (PHB/5ChNC has a Tc of 49 °C). Figure 6b, which relates to the second heating cycle, shows that the cold crystallization peak of the nanocomposites is at approximately 42 °C (Tcc); this peak is not observed for PHB. Cold crystallization is typical in polymers with low crystallinity and is related to the difficulty of the polymer chains to crystallize during the cooling phase from the melt. Moreover, as in the first heating cycle, a lower degree of crystallinity was observed in the presence of ChNCs. These results, together with the lower Tc shown in Figure 6b and discussed previously, indicate that the presence of ChNCs hinders the crystallization process both in terms of the crystallization rate and degree of crystallinity. These results agree with the POM study. The reduced crystallinity of the nanocomposites may be attributed to the hindered motion of polymer segments caused by the presence of ChNCs in the matrix. Notably, the cold crystallization peak appeared only after a phase of controlled cooling at 10 °C/min, as described in the DSC analysis. This peak is not visible during the first heating cycle, which means that the cooling phase during sheet extrusion was slower and allowed full crystallization of the nanocomposites from the melt. According to the literature [53], in some cases, the presence of ChNCs may decrease the crystallization degree of PHB because of a reduction in the lamellar thickness of PHB, resulting in confined PHB molecules in the blends. Finally, the melting temperature (Tm) did not change when ChNCs were added to the PHB-based formulations. Notably, however, some variations in the shape of the melting peak were observed, particularly in the second heating cycle. Neat PHB showed two distinct melting peaks: a lower peak at approximately 160 °C and an upper peak at approximately 170 °C. The upper melting peak is dominant, suggesting that the PHB sample has a nonhomogeneous morphology with mostly stable crystals, but unstable crystals are also present and agrees with the results obtained by Owen et al. [54]. However, the nanocomposites showed only one melting peak at 170 °C, with a shoulder at 160 °C, suggesting that a higher fraction of PHB crystals are present in their stable configuration, and a more homogeneous morphology is induced by the ChNCs.

Figure 7 shows the complex viscosity η* of neat PHB and its nanocomposites as a function of the oscillation frequency. PHB shows shear-thinning behavior, with the viscosity decreasing with increasing frequency. This behavior is more pronounced in the low-frequency region. The addition of ChNCs increased the viscosity, and the nanocomposite with the highest ChNC content exhibited the highest viscosity. This viscosity trend was consistently observed in the entire studied frequency range.

Figure 8 shows representative stress-strain curves of neat PHB and its nanocomposites in the machine direction (MD) and transverse direction (TD), and the results are reported in Table 2. The graphs in Figure 8 show that the mechanical properties in MD are significantly higher than those in TD. This effect is evident even in the neat PHB. This is due to the calendering process, which is resulting in the alignment of the polymer at the molecular level and is previously reported for polymer and biopolymer calendered films [55]. Furthermore, the addition of ChNCs significantly enhances the mechanical strength of the PHB. A substantial increase in the tensile strength, from 24 MPa for neat PHB to 27 MPa (1 wt.% ChNCs) to 33 MPa (5 wt.% ChNCs), corresponding to an increase of 17% and 38%, respectively, is seen in MD. The samples in TD exhibit a similar pattern and suggest an improvement in tensile strength due to the presence of ChNCs, although the strength values remain lower than the MD one, this is explained with lack of orientation of the molecule chains in the transversal direction. The improved tensile strength is an indication of a good interface between PHB and the ChNCs. The addition of ChNCs does not significantly affect the tensile modulus in MD, while a slight decrease is seen in TD. All the materials displayed brittle behavior with a low elongation at break. Interestingly, the strain at the break did not decrease with the addition of agglomerated ChNCs, which is a common effect of good stress transfer [31,56,57].

Figure 9 shows the microstructures of PHB and its nanocomposite sheets. The neat PHB sheet in Figure 9a is homogeneous, whereas multiple particles are observed in both nanocomposites in Figure 9b,c, and the number of particles increases with the ChNC concentration. The ChNC material structure is very similar (a flake-like structure), as previously reported by Herrera et al. [58]. Therefore, these particles are most likely ChNCs agglomerates that were not dispersed at the nanoscale.

Figure 10 shows the fractured surfaces of neat PHB and its nanocomposites with 1 and 5 wt.% ChNC contents. These micrographs indicate that all the samples underwent brittle fracture, which agrees with the results of mechanical testing. The fractured surface structure of the nanocomposites looks quite different from that of neat PHB (Figure 10a), and the difference is especially prominent for the nanocomposite with the lower ChNCs content (Figure 10b). However, both nanocomposites show micrometer-sized, layered, flake-like structures (Figure 10b,c). In addition, the nanocomposite with 5 wt.% ChNC has several agglomerates that are larger than those in the nanocomposite with 1 wt.% ChNCs. These results agree with the OM analysis of the sheets. The formation of these ChNC agglomerates can be attributed to the freeze-drying step before the compounding process. The ChNCs form hydrogen bonds, resulting in an irreversible agglomeration that cannot be redispersed in the PHB matrix during the compounding process.

## 4. Conclusions

This study reports the properties of PHB biopolymer reinforced with ChNCs. The PHB and ChNCs used in this study were produced at a large scale from biomass and chitin powder, respectively. The materials (neat PHB and nanocomposites) were prepared by melt extrusion, and the materials for testing were prepared by calendering.

We successfully demonstrated the large-scale production of ChNCs using H_2_SO_4_ acid hydrolysis. However, the freeze-drying step to facilitate the transport of large ChNCs resulted in irreversible agglomeration, as shown in the OM images.

Polarized optical microscopy showed that pure PHB had a higher crystallization rate and formed larger spherulites than the nanocomposites. The DSC results showed that the presence of ChNCs resulted in a lower degree of crystallinity and a slower crystallization rate. However, some variations in the shape of the melting peak were observed for the nanocomposites, presenting only one melting peak at 170 °C with a shoulder at 160 °C. It is assumed that the presence of ChNCs induced the formation of a higher fraction of more stable PHB crystals and a more homogeneous nanocomposite morphology.

The addition of ChNCs significantly enhanced the mechanical strength of the nanocomposites in both the machine (MD) and transversal directions (TD). The samples in the MD showed significantly higher mechanical properties than those in the TD direction, owing to the partial orientation during the calendering process.

Morphological studies performed by OM and SEM showed that despite the presence of a few larger micro-sized agglomerates of ChNCs due to their dry feeding during the melt mixing process, a uniform distribution was obtained.

This study has demonstrated that adding a small amount of bio-based biodegradable nanoparticles, such as ChNCs, has a toughening effect on PHB, which is well known to exhibit brittle behavior and can therefore promote the use of this material even in applications where good mechanical properties are required. It is worth noting that these results were obtained by mixing ChNCs in a polymer matrix via melt extrusion in a twin-screw extruder, which is a typical process commercially used with thermoplastics; therefore, these results can be easily scaled up to the industrial level.

## Figures and Tables

**Figure 1 polymers-14-00562-f001:**
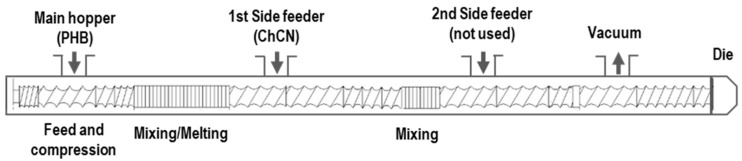
The layout of the twin-screw extruder and its screw design.

**Figure 2 polymers-14-00562-f002:**
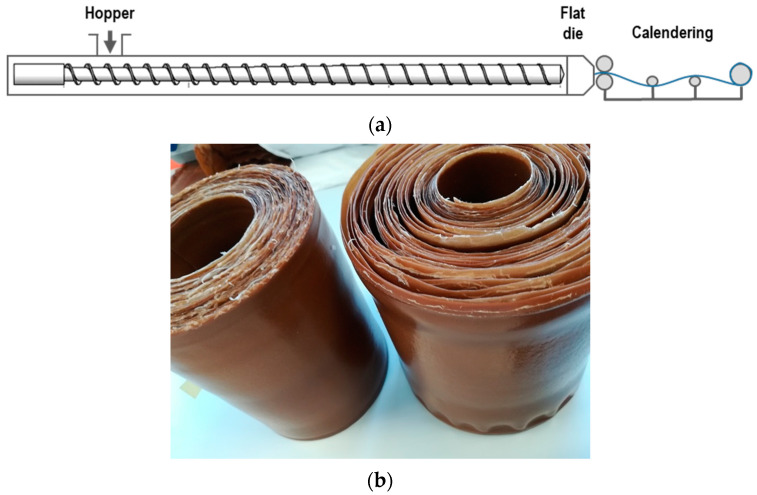
(**a**) Film extrusion and calendering process layout and (**b**) calendered sheets.

**Figure 3 polymers-14-00562-f003:**
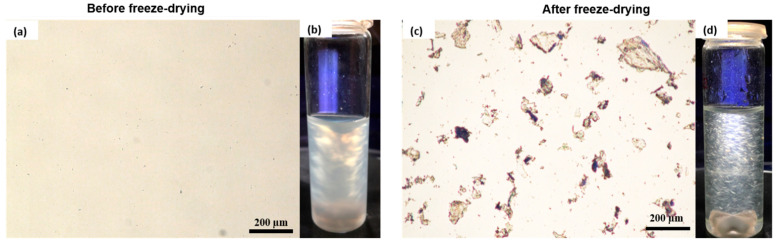
Optical micrographs of ChNCs (**a**) before and (**c**) after freeze-drying steps and the birefringence of ChNCs in water suspensions (**b**) before and (**d**) after freeze-drying with similar concentrations.

**Figure 4 polymers-14-00562-f004:**
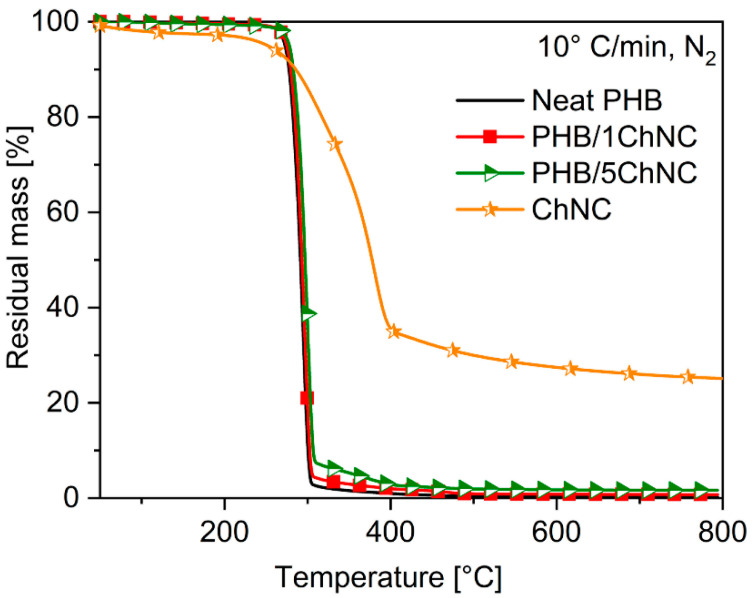
Weight loss as a function of temperature for ChNCs, PHB, and the ChNC-nanocomposites of PHB in a nitrogen atmosphere.

**Figure 5 polymers-14-00562-f005:**
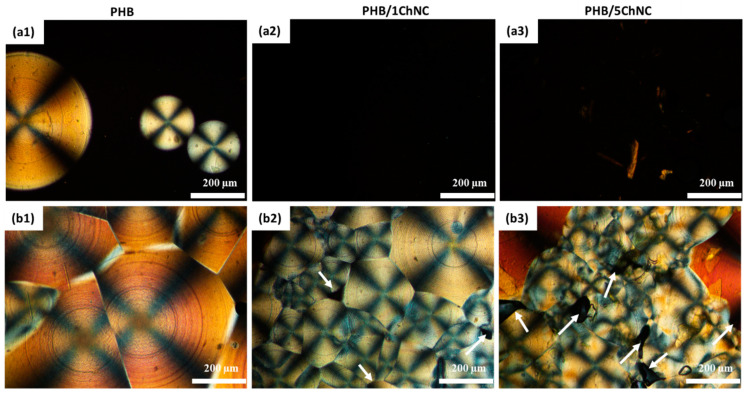
Polarized optical micrographs of PHB, PHB/1ChNC, and PHB/5ChNC at 80 °C for 1 min are shown in (**a1**–**a3**), respectively and at 60 °C for 2 min are shown in (**b1**–**b3**), respectively.

**Figure 6 polymers-14-00562-f006:**
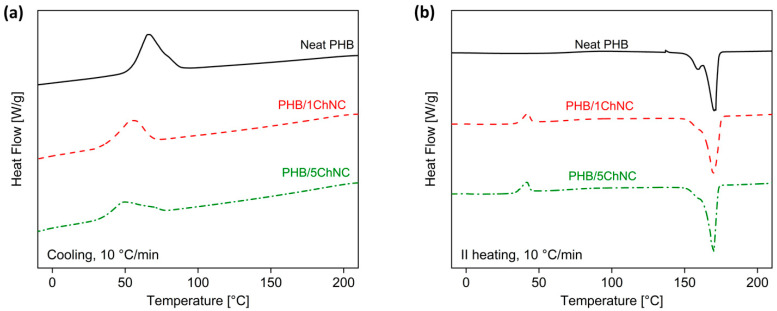
Thermal properties of neat PHB compared with the nanocomposites: (**a**) cooling cycle from 210 to −10 °C, (**b**) second heating cycle from −10 to 210 °C.

**Figure 7 polymers-14-00562-f007:**
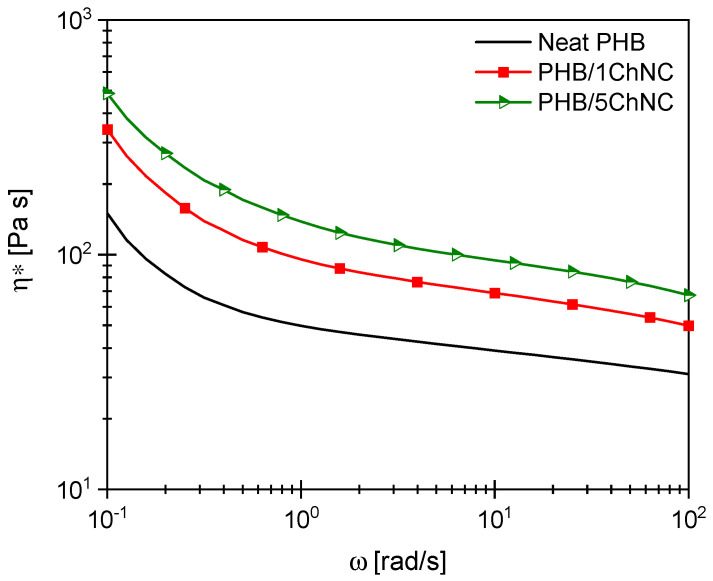
Rheology of the studied PHB-based nanocomposites.

**Figure 8 polymers-14-00562-f008:**
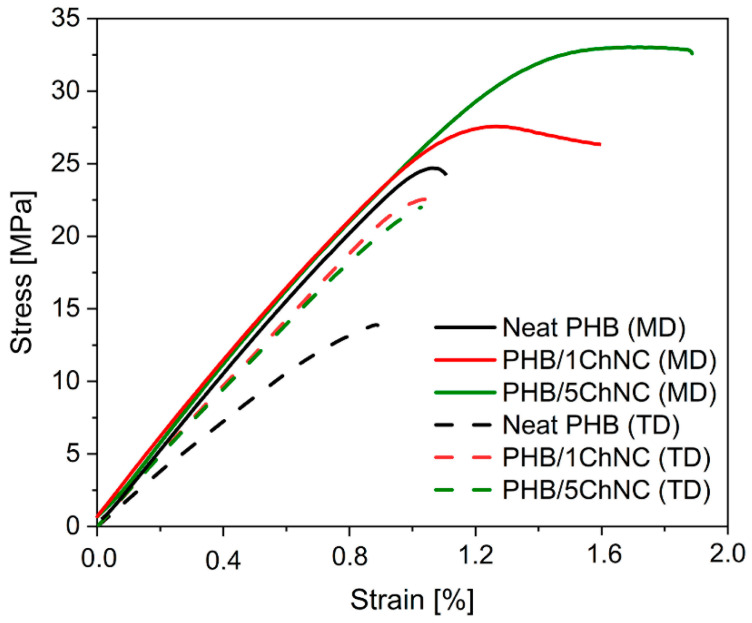
Tensile stress-strain curves of PHB and its nanocomposites with ChNCs in MD and TD directions.

**Figure 9 polymers-14-00562-f009:**
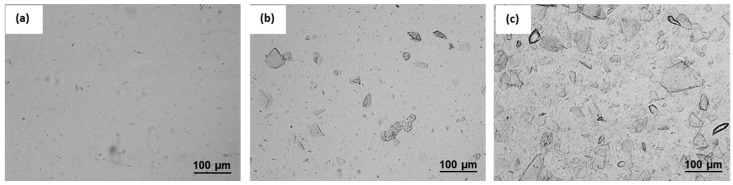
The OM images of cast-extruded (**a**) PHB, (**b**) PHB/1ChNC, and (**c**) PHB/5ChNC show an increasing number of ChNC particles in the PHB matrix with increasing ChNC content.

**Figure 10 polymers-14-00562-f010:**
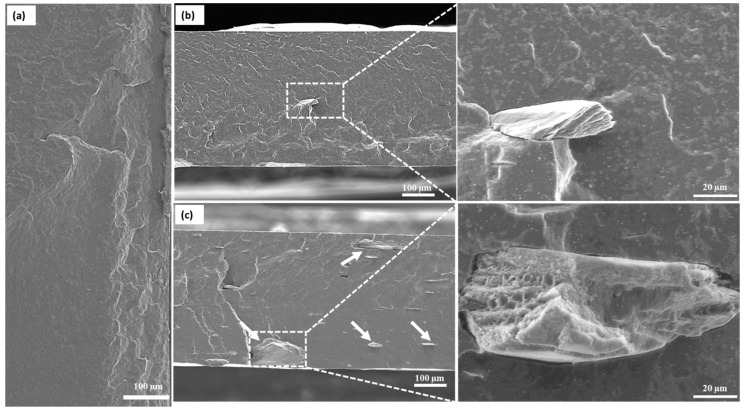
SEM images of the cryo-fractured surfaces of (**a**) PHB, (**b**) PHB/1ChNC, and (**c**) PHB/5ChNC.

**Table 1 polymers-14-00562-t001:** DSC results of the studied nanocomposites. Values for each cycle are reported.

Materials	Cooling		2nd Heating				
	T_c_ [°C]	ΔH_c_ [J/g]	Tcc [°C]	ΔH_cc_ [J/g]	T_m_ [°C]	ΔH_m_ [J/g]	X_c_ [%]
Neat PHB	66	58	-	0	170	88	60
PHB/1ChNC	54	36	42	8	170	82	50
PHB/5ChNC	49	29	41	10	170	79	49

**Table 2 polymers-14-00562-t002:** Tensile test results in both MD and TD for the studied nanocomposites and neat PHB.

Materials		E-Modulus [GPa]	σ_max_ [MPa]	ε at σ_max_ [%]	ε_break_ [%]	Toughness [MJ/m^3^]
PHB	MD	2.7 ± 0.2	24 ± 2	1.1 ± 0.1	1.2 ± 0.1	0.13 ± 0.0
	TD	1.8 ± 0.2	14 ± 0	0.9 ± 0.2	1.0 ± 0.2	0.06 ± 0.1
PHB/1ChNC	MD	2.7 ± 0.1	27 ± 1	1.3 ± 0.1	1.5 ± 0.2	0.18 ± 0.1
	TD	2.4 ± 0.3	22 ± 0	1.1 ± 0.1	1.1 ± 0.1	0.13 ± 0.1
PHB/5ChNC	MD	2.8 ± 0.1	33 ± 4	1.6 ± 0.4	1.7 ± 0.4	0.22 ± 0.0
	TD	2.5 ± 0.2	21 ± 0	1.0 ± 0.1	1.0 ± 0.1	0.12 ± 0.0

## Data Availability

Data is available on request.

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
