# Peer review of "Influence of Chitin Nanocrystals on the Crystallinity and Mechanical Properties of Poly(hydroxybutyrate) Biopolymer"

_polymers, 2022, doi:10.3390/polym14030562_

Round 1

Reviewer 1 Report

Manuscript polymers-1551199 describe the production and the characterization of the polyhydroxy butyrate (PHB) and chitin nanocrystals (ChNCs) nanocomposites. The production was done at the pilot scale. The nanocomposites were compounded using a twin-screw extruder. This preparation method is a novelty of the manuscript, and it represents a viable solution to scale up the production of PHB-ChNCs nanocomposites. Other papers present the preparation of nanochitin/polyhdroxybutyrate nanocomposites by different approaches, which are not very easy to scale up, such as Pickering emulsions (Baraki et al. 2021,  Composites Communications, 25, 100655) or casting from a solution (Reddy et al. 2015,  Polymer, 75, 141-150, Ikejima et al. 1999, Macromol. Chem. Phys., 200, 413-421). The authors underlined in the Conclusion section this novelty.

General comments.

The manuscript fits in the Polymers journal aim and scope and is well-written. However, before publication, the manuscript needs significant improvements. The main weakness of the manuscript is related to the lack of characterization of the used chitin and the lack of statistical analysis related to the intrinsic variability of chitin. The chitin supplier, Glentham Life Sciences,  Corsham, UK, re-sales (smaller quantity of) chitin, is provided by several industrial producers. Chitin is a natural polymer, highly variable, produced at an industrial level both from (marine) crustacean shells and fungal (mushroom) biomass. Chitin main characteristics (molecular mass, type of allomorphic crystal) depend mainly on the source of chitin and the extraction method. There are three crystalline allomorphic types of chitin - α-chitin, β-chitin, and γ-chitin. The authors did not mention the source of chitin, the molecular mass, and the allomorphic type. Considering the high variability of the chitin that the authors used as a raw material to prepare the chitin nanocrystals (ChNCs), it is necessary to perform a statistical analysis based on different batches of chitin (from the same source, marine or fungal). The crystallinity of the prepared nano-chitin must also be investigated by  X-ray diffraction. From this investigation, the chitin nanocrystals dimensions could also be determined. The corroborating evidence is essential for a scientific basis of process development. Different batches of chitin could also influence the sulfation degrees. Statistical analysis must be done for ChNCs crystallinity, nanoparticles dimension, and sulfation degrees (determined as sulfate ash in the thermogravimetric analysis). Such statistical analysis should ensure the PHB /ChNCs nanocomposites production process reproducibly.

Specific comments.

L322. “Figure 8. Tensile stress-strain curves of PHB and its nanocomposites with ChNCs: (a) MD and (b). TD” There are no part (a) and part (b) of Figure 8, both MD and TD being represented in the same graph.

L335. Error! Reference source not found! Please correct.

Author Response

Reviewer 1: Manuscript polymers-1551199 describe the production and the characterization of the polyhydroxy butyrate (PHB) and chitin nanocrystals (ChNCs) nanocomposites. The production was done at the pilot scale. The nanocomposites were compounded using a twin-screw extruder. This preparation method is a novelty of the manuscript, and it represents a viable solution to scale up the production of PHB-ChNCs nanocomposites. Other papers present the preparation of nanochitin/polyhdroxybutyrate nanocomposites by different approaches, which are not very easy to scale up, such as Pickering emulsions (Baraki et al. 2021, Composites Communications, 25, 100655) or casting from a solution (Reddy et al. 2015, Polymer, 75, 141-150, Ikejima et al. 1999, Macromol. Chem. Phys., 200, 413-421). The authors underlined in the Conclusion section this novelty.

Respected reviewer, thank you so much for your feedback on our work. We are very grateful for the time you have given to the study and hope that the revisions we have made are satisfactory. The following text includes our point-by-point responses to the reviewer’s comments.

General comments.

The manuscript fits in the Polymers journal aim and scope and is well-written. However, before publication, the manuscript needs significant improvements. The main weakness of the manuscript is related to the lack of characterization of the used chitin and the lack of statistical analysis related to the intrinsic variability of chitin. The chitin supplier, Glentham Life Sciences, Corsham, UK, re-sales (smaller quantity of) chitin, is provided by several industrial producers. Chitin is a natural polymer, highly variable, produced at an industrial level both from (marine) crustacean shells and fungal (mushroom) biomass. Chitin's main characteristics (molecular mass, type of allomorphic crystal) depend mainly on the source of chitin and the extraction method. There are three crystalline allomorphic types of chitin - α-chitin, β-chitin, and γ-chitin. The authors did not mention the source of chitin, the molecular mass, and the allomorphic type. Considering the high variability of the chitin that the authors used as a raw material to prepare the chitin nanocrystals (ChNCs), it is necessary to perform a statistical analysis based on different batches of chitin (from the same source, marine or fungal). The crystallinity of the prepared nano-chitin must also be investigated by  X-ray diffraction. From this investigation, the chitin nanocrystals dimensions could also be determined. The corroborating evidence is essential for a scientific basis of process development. Different batches of chitin could also influence the sulfation degrees. Statistical analysis must be done for ChNCs crystallinity, nanoparticles dimension, and sulfation degrees (determined as sulfate ash in the thermogravimetric analysis). Such statistical analysis should ensure the PHB /ChNCs nanocomposites production process reproducibly.

Response: Thank you so much for your valuable comments on chitin and we agree with your observations. The chitin powder from Glentham Life Sciences (code: GC0425) is derived from shrimp shell origin. The molecular mass of chitin, as well as its origin, have now been included in the material section.

The chitin isolated from shrimp shells (crustaceans), has an α-form crystalline structure, where the crystals are aligned in an anti-parallel structure, resulting in a strong and rigid exterior shell. Also, the XRD analysis of chitin powder confirms the presence of α-form crystalline structure, which is added as Figure S5 in the supplementary information. Regarding your comment on statistical analysis,  we agree that would be needed if the study was focusing on different chitin sources and isolation processes. In this work, we have used only one type of chitin source and one process (one batch) and therefore we do not have any statistical analysis. The chitin nanocrystals (ChNCs) used in this study were characterized for their thermal properties, microstructure, birefringence as well as size, the size ranges of the nanocrystals is presented in the supplementary information. We would also like to point out that ChNCs are not commercially available, and the pilot-scale isolation made in this work is not reported before. Furthermore, for this work, the most important characteristics of the used ChNCs were their thermal stability, birefringence, and size, to ensure that the ChNCs were isolated to nanosize and had thermal stability necessary for the high-temperature extrusion and calendering processes.

Specific comments.

  1. “Figure 8. Tensile stress-strain curves of PHB and its nanocomposites with ChNCs: (a) MD and (b). TD” There are no part (a) and part (b) of Figure 8, both MD and TD being represented in the same graph.

Response: Thanks so much, the caption of Figure 8 has been modified with the sentence “Figure 8. Tensile stress-strain curves of PHB and its nanocomposites with ChNCs in MD and TD directions.”

  1. Error! Reference source not found! Please correct.

Response: Thank you for noticing this error; the sentence has now been corrected and the link was removed.

Reviewer 2 Report

The topic of the work is very interesting and fits very well with the current research trend concerning biopolymers and their composites.

The work is very well prepared, well written and with very interesting results.

Both the introduction, methodology, discussion of the obtained results and conclusions are written correctly, I have no significant substantive comments to them.

English requires improvement and punctuation in some places.

Additional remarks:

text and Table 1 - units, please write in italics

Tensile properties - please explain why the tensile strengths of composite samples prepared in the direction parallel to the machine axis (MD) were greater than for materials from the transverse direction?

With that, my decision is minor review.

Author Response

Reviewer 2: The topic of the work is very interesting and fits very well with the current research trend concerning biopolymers and their composites. The work is very well prepared, well written, and with very interesting results. Both the introduction, methodology, discussion of the obtained results, and conclusions are written correctly, I have no significant substantive comments to them.

Thank you very much for your positive and constructive feedback. We are glad to submit a revised version after addressing all the comments. We hope that the revisions we have made are satisfactory.

  1. English requires improvement and punctuation in some places.

Response: Dear Reviewer, this manuscript was edited by a professional proofreading service before the submission to the journal. We can provide the.certificate if needed. Howevere, we have  found some  mistakes which have been corrected.

Additional remarks: text and Table 1 - units, please write in italics

Response: Thank you for the suggestion. Units are now been modified in italics in Table 1 and Table 2.

Tensile properties - please explain why the tensile strengths of composite samples prepared in the direction parallel to the machine axis (MD) were greater than for materials from the transverse direction?

Response: The mechanical properties in the MD and TD are different because of the molecular orientation of the polymer in MD. This orientation is especially seen in the tensile strength of the neat PHB and its nanocomposites. Calendering or film blowing processes are typically leading to polymer orientation and therefore also a difference in strength. We have revised this and explained this better in the manuscript.

With that, my decision is a minor review.

Response: Thanks so much.

Round 2

Reviewer 1 Report

Authors made the requested improvements and upgrade the manuscript.